# Prescribed Performance-Based Consensus Tracking for Multiagent Systems*

Yihang Chen
*School of Automation Engineering*
*University of Electronic Science and Technology of China*
Chengdu, China
yihangchen2022@163.com

*Abstract*—This paper investigates the prescribed performance-based cooperative control problem for a class of uncertain strict-feedback nonlinear multiagent systems with input saturation. Compared with the existing results, the new performance function is constructed with a larger initial value and faster convergence speed to meet the feasible condition of prescribed performance control, while ensuring that tracking errors converge to the prescribed range in a configurable time frame. On this basis, the constrained system is converted into an equivalent totally unconstrained one, which indicates that the convergence of the logarithmic error transformation function is mapped to the convergence of the tracking error. Meanwhile, the problem of input saturation is considered, which can be addressed by constructing an auxiliary system with the same order as the dynamics for compensating the effect of input saturation. Moreover, for the topology communication redundancy problem, a topology optimization scheme is designed to eliminate unnecessary edges and save communication resources. The results of numerical simulation on the control systems verify the approach.

*Index Terms*—cooperative control, prescribed performance control, input saturation, topology optimization.

## I. Introduction

**D**URING the past few decades, people gradually pay attention to the research on multiagent systems (MASs) due to the ability to attack more complex, realistic, and large-scale problems which are beyond the capabilities of an individual agent, such as extensive search and rescue [1], traffic vehicle [2], environmental monitor [3]. A research goal is to transform large complex systems (hardware and software systems) into easy-to-manage systems in which individual agents can communicate and coordinate with each other and eventually complete control objective precisely. Accomplishing precise control has always been an essential control requirement for the smooth operation of multiagent systems. Researchers have achieved abundant results; see, for instance [4]. The control outcomes of formation control, consensus control, and containment control are commonly demonstrated by the convergence of tracking errors to a small vicinity around the origin. Extensive investigation has been undertaken regarding the design of stable controllers especially for systems with nonlinear uncertain terms, including fuzzy systems [5], adaptive backstepping [6], adaptive feedback [7].

The purpose of this paper is to reconstruct the performance function for a particular class of tracking error initial values that exceed its boundaries. The reconstruction aims to meet the following requirements

1) The initial value $\rho_i(0)$ of the new performance function satisfies $\rho_i(0) > |e_i(0)|$, where $e_i(0)$ represents the error initial value.

2) The new prescribed performance function should converge within the boundaries of the original performance function after a designable time $\tau_{mp}$.

With the above observations, this work investigates consensus tracking problem of multiagent systems via a performance function reconstruction approach and topology optimization method. The advancements of this paper are delineated as follows

1) A new performance function is designed to satisfy the feasible condition of PPC and larger convergence speed of tracking errors. Compared to the performance function delineated in [8], the new performance function expedites the convergence rate of the tracking error and mitigates tracking error overshoots by incorporating a specifically designed logarithmic term.

2) A new topology optimization method is developed based on the minimum communication distance, which can eliminate redundant edges while ensuring that the communication graph contains a spanning tree.

The structure of the remaining sections of this paper is as follows. Nonlinear dynamics model and some preliminaries including graph theory, fuzzy logic system, topology optimization as well as prescribed performance scheme are introduced in Section II. Controller and adaptive laws based backstepping approach are designed in Section III. Numerical examples are provided in provided in Section IV.

## II. Problem Formulation and Preliminaries

The dynamics of the $i$th $(i = 1, ..., N)$ follower is described in the strict-feedback form

$$
\begin{aligned}
\dot{x}_{i,m} &= x_{i,m+1} + f_{i,m}(\bar{x}_{i,m}) + \omega_{i,m}(t), \\
\dot{x}_{i,n} &= u_i(v_i(t)) + f_{i,n}(\bar{x}_{i,n}) + \omega_{i,n}(t), \\
y_i &= x_{i,1}, \quad m = 1, 2, ..., n-1
\end{aligned}
\tag{1}
$$

where $x_i = [x_{i,1}, ..., x_{i,n}]^T \in \mathbb{R}^n$ and $y_i$ represent state vector and system output of the $i$th agent with $\bar{x}_{i,k} =$

$[x_{i,1}, ..., x_{i,k}]^T \in \mathbb{R}^k(k = 1, ..., n)$, $f_{i,k}(\bar{x}_{i,k})$ is unknown smooth nonlinear function, $\omega_{i,k}(t)$ is unknown time-varying disturbance, The system input signal $u_i(v_i(t))$ is affected by saturation nonlinearity, as expressed by

$$u_i(v_i(t)) = \begin{cases} sign(v_i(t))u_{i,M}, & |v_i(t)| \geq u_{i,M} \\ v_i(t), & |v_i(t)| < u_{i,M} \end{cases} \quad (2)$$

where $u_{i,M} > 0$ represents the known upper bounds of $u_i(v_i(t))$, and $v_i(t)$ denotes the actuator input to be designed.

The leader's signal $y_r$ is assumed to be generated by a linear autonomous system of the form

$$\dot{x}_l = Sx_l,$$
$$y_r = x_{l,1}, \quad (3)$$

where $x_l = [x_{l,1}, ..., x_{l,n}]^T \in \mathbb{R}^n$ is the state vector of leader node, $S \in \mathbb{R}^{n \times n}$ is the constant coefficient matrix, we assume that $S$ is detectable.

### A. Graph Theory

Consider the direct graph $\Upsilon = (\mathcal{U}, \mathcal{E}, \mathcal{P})$, where $\mathcal{U} = \{1, ..., N\}$ are the sets of nodes, $\mathcal{E} = \{(i,j) : i \in \mathcal{U}, j \in \mathcal{U}\}$ are edges between $i$th agent and $j$th agent, $\mathcal{P} = [a_{i,j}]_{N \times N}$ stands for the adjacency matrix. If $a_{i,j} = 0$, there is no information interaction between $i$th and $j$th agents; otherwise $a_{i,j} > 0$. The Laplacian matrix is defined as $\mathcal{L} = \mathcal{D} - \mathcal{P}$, where $\mathcal{D} = \text{diag}\{d_1, ..., d_N\}$ with $d_i = \sum_{j=1}^N a_{i,j}$ is the in-degree matrix of the $i$th follower. In order to achieving subsequent study, the following assumptions are imposed.

The augmented graph is denoted as $\bar{\Upsilon} = (\bar{\mathcal{U}}, \bar{\mathcal{E}})$ with $\bar{\mathcal{U}} = \{0, 1, ..., N\}$ and $\bar{\mathcal{E}} = \{(i,j) : i \in \bar{\mathcal{U}}, j \in \mathcal{U}\}$, $a_{i,0} > 0$ represent $i$th agent maintains communication with the leader, $\mathcal{A} = \text{diag}\{a_{1,0}, a_{2,0}, ..., a_{N,0}\} \in \mathbb{R}^{N \times N}$.

### B. Fuzzy Logic System

In order to approximate the unknown continue function, we introduce the following definitions and lemma. Consider the fuzzy rule base consisting of $M$ rules in the following form

$$R_j : \text{If } \text{x}_1 \text{ is } A_1^j, \text{ x}_2 \text{ is } A_2^j, ..., \text{ x}_n \text{ is } A_n^j,$$
$$\text{Then } z \text{ is } B^j,$$

with $j = 1, 2, ..., M$, $\text{X} = [\text{x}_1, \text{x}_2, ..., \text{x}_n]^T \in \mathbb{R}^n$ and $z$ as the input vector and the output variable of the fuzzy system, respectively. $A_i^j$ and $B^j$ represent linguistic terms defined by fuzzy membership functions $\mu_{A_i^j}(\text{x}_i)$ and $\mu_{B^j}(z)$, respectively.

The fuzzy logic system has the form

$$g(\text{X}) = \frac{\sum_{j=1}^M \bar{z}^j \Pi_{i=1}^n \mu_{A_i^j}(\text{x}_i)}{\sum_{j=1}^M [\Pi_{i=1}^n \mu_{A_i^j}(\text{x}_i)]}, \quad (4)$$

where $\mu_{A_i^j}(\text{x}_i)$ is the Gaussian membership function, denoted as

$$\mu_{A_i^j}(\text{x}_i) = a_i^j \exp\left[-\frac{1}{2}\left(\frac{\text{x}_i - \bar{\text{x}}_i^j}{\sigma_i^j}\right)^2\right], \quad (5)$$

herein, $a_i^j$, $\bar{\text{x}}_i^j$, and $\sigma_i^j$ are real-valued parameters with $0 < a_i^j \leq 1$ and $\bar{z}^j = \max_{z \in R} \mu_{B^j}(z)$.

### C. Prescribed Performance

Throughout this work, the cooperative control objective is to confine $e_i(t) = \sum_{j=1}^N a_{i,j}(y_i - y_j) + a_{i,0}(y_i - y_r)$ within a predefined arbitrarily small residual set.

$$-\rho_i(t) < e_i(t) < \rho_i(t), \quad (6)$$

where the candidate function could be $\rho_i(t) = (\rho_i(0) - \rho_i(\infty))e^{-lt} + \rho_i(\infty)$ which is a strictly positive and monotonically decreasing function satisfying $\rho_i(0) > \lim_{t \to \infty} \rho_i(t) > 0$, $l > 0$ is a constant representing the required exponential convergence rate. Clearly, $\rho_i(t)$ is bounded and continuously differentiable. $\rho_i(0)$ is the initial performance bound, $\rho_i(\infty)$ is the maximum steady state error.

The prerequisite for achieving the above control objective is to select suitable initial value $\rho_i(0)$ and satisfy $\rho_i(0) > |e_i(0)|$. We consider the case that the precise value of initial error $e_i(0)$ can be obtained, a performance function $\rho_i^*(t)$ is expressed by

$$\rho_i^*(t) = (\rho_i^*(0) - \rho_i(\infty)) \exp\left[-(l - \frac{\ln \frac{\rho_i(0) - \rho_i(\infty)}{\rho_i^*(0) - \rho_i(\infty)}}{\tau_{mp}})t\right] + \rho_i(\infty), \quad (7)$$

where $\rho_i^*(0) = \hbar|e_i(0)|$ with $\hbar > 1$ is a constant, $\tau_{mp} > 0$ is an arbitrarily specified constant which satisfies $\rho_i^*(\tau_{mp}) = \rho_i(\tau_{mp})$. Then, a new performance function is designed as

$$\mathcal{H}_i(t) = \begin{cases} \rho_i(t), & |e_i(0)| < \rho_i(0) \\ \rho_i^*(t). & |e_i(0)| \geq \rho_i(0) \end{cases} \quad (8)$$

## III. MAIN RESULTS

**Step $m(1 \leq m \leq n-1)$:** The time derivative of $z_{i,m}$ can be denoted as

$$\dot{z}_{i,m} = \dot{x}_{i,m} - \dot{\varpi}_{i,m-1} - \dot{\varsigma}_{i,m}$$
$$= x_{i,m+1} + f_{i,m} + \omega_{i,m} - \dot{\varpi}_{i,m-1} - \varsigma_{i,m+1} + p_{i,m}\varsigma_{i,m}$$
$$= z_{i,m+1} + \varpi_{i,m} + f_{i,m} + \omega_{i,m} - \dot{\varpi}_{i,m-1} + p_{i,m}\varsigma_{i,m}. \quad (9)$$

where $\dot{\varpi}_{i,m-1}$ is denoted as

Define the following Lyapunov function:

$$V_{i,m} = \frac{1}{2}z_{i,m}^2 + \frac{1}{2\kappa_{i,m}}\tilde{\vartheta}_{i,m}^2 + \frac{1}{2\gamma_{i,m}}\tilde{\Gamma}_{i,m}^2, \quad (10)$$

where $\kappa_{i,m} > 0$ and $\gamma_{i,m} > 0$ are designed constant, $\tilde{\vartheta}_{i,m} = \vartheta_{i,m} - \hat{\vartheta}_{i,m}$ and $\tilde{\Gamma}_{i,m} = \Gamma_{i,m} - \hat{\Gamma}_{i,m}$ are the estimation errors. Then, the derivative of $V_{i,m}$ along (9) is

$$\dot{V}_{i,m} = z_{i,m}(z_{i,m+1} + \varpi_{i,m} + f_{i,m} + \omega_{i,m} - \dot{\varpi}_{i,m-1}$$
$$+ p_{i,m}\varsigma_{i,m}) - \frac{1}{\kappa_{i,m}}\tilde{\vartheta}_{i,m}\dot{\hat{\vartheta}}_{i,m} - \frac{1}{\gamma_{i,m}}\tilde{\Gamma}_{i,m}\dot{\hat{\Gamma}}_{i,m}, \quad (11)$$

given that $\bar{f}_{i,m} = f_{i,m} - \dot{\varpi}_{i,m-1}$ is an unknown function, a fuzzy logic system $W_{i,m}^T \varphi_{i,m}(\text{X}_{i,m})$ is utilized to approximate $\bar{f}_{i,m}$ with $\text{X}_{i,m} = [\bar{x}_{i,m}^T, \bar{x}_{j,m}^T, \beta_i^{(0)}, ..., \beta_i^{(m-1)}, \hat{\vartheta}_{i,1}, ..., \hat{\vartheta}_{i,m-1}, \hat{\Gamma}_{i,1}, ..., \hat{\Gamma}_{i,m-1}, \varsigma_{i,1}, ..., \varsigma_{i,m-1}, y_r]^T$. Thus, $\bar{f}_{i,m}$ can be expressed as

$$\bar{f}_{i,m} = W_{i,m}^T \varphi_{i,m}(\text{X}_{i,m}) + \delta_{i,m}, \quad |\delta_{i,m}| \leq \varepsilon_{i,m}. \quad (12)$$

Define $\Gamma_{i,m} = \varepsilon_{i,m} + \bar{\omega}_{i,m}, \vartheta_{i,m} = ||W_{i,m}||$, design the adaptive laws as

$$\dot{\hat{\vartheta}}_{i,m} = \kappa_{i,m}\frac{z_{i,m}^2\varphi_{i,m}^T\varphi_{i,m}}{\sqrt{z_{i,m}^2\varphi_{i,m}^T\varphi_{i,m} + \sigma_{i,m}^2}} - \kappa_{i,m}\hat{\vartheta}_{i,m}, \quad (13)$$

$$\dot{\hat{\Gamma}}_{i,m} = \gamma_{i,m}\frac{z_{i,m}^2}{\sqrt{z_{i,m}^2 + \sigma_{i,m}^2}} - \gamma_{i,m}\hat{\Gamma}_{i,m}, \quad (14)$$

where $\sigma_{i,m} > 0$. Combining Young's inequality, substituting (12)-(14) into (11) yields

$$\dot{V}_{i,m} \leq z_{i,m}(z_{i,m+1} + \varpi_{i,m} + p_{i,m}\varsigma_{i,m}) + \frac{\hat{\Gamma}_{i,m}z_{i,m}^2}{\sqrt{z_{i,m}^2 + \sigma_{i,m}^2}}$$
$$+ \frac{\hat{\vartheta}_{i,m}z_{i,1}^2\varphi_{i,m}^T\varphi_{i,m}}{\sqrt{z_{i,m}^2\varphi_{i,m}^T\varphi_{i,m} + \sigma_{i,m}^2}} - \frac{1}{2}(\tilde{\vartheta}_{i,m}^2 + \tilde{\Gamma}_{i,m}^2)$$
$$+ \frac{1}{2}(\vartheta_{i,m}^2 + \Gamma_{i,m}^2) + \sigma_{i,m}(\vartheta_{i,m} + \Gamma_{i,m}). \quad (15)$$

The virtual controller $\varpi_{i,m}$ is designed as

$$\varpi_{i,m} = - c_{i,m}z_{i,m} - \Xi_{i,m}z_{i,m-1} - \frac{\hat{\vartheta}_{i,m}z_{i,m}\varphi_{i,m}^T\varphi_{i,m}}{\sqrt{z_{i,m}^2\varphi_{i,m}^T\varphi_{i,m} + \sigma_{i,m}^2}}$$
$$- \frac{\hat{\Gamma}_{i,m}z_{i,m}}{\sqrt{z_{i,m}^2 + \sigma_{i,m}^2}} - p_{i,m}\varsigma_{i,m}, \quad (16)$$

it is interesting to note that $\Xi_{i,m}$ exhibits the following properties: 1) $\Xi_{i,m} = \lambda_{i,1}(d_i + a_{i,0})$ if and only if $m = 2$; 2) $\Xi_{i,m} = 1$ as $m = 3, 4, ..., n - 1$ and $c_{i,m} > 0$ is a constant.

Substituting (12)-(16) into (11) yields

$$\dot{V}_{i,m} \leq z_{i,m}z_{i,m+1} - c_{i,m}z_{i,m}^2 - \Xi_{i,m}z_{i,m}z_{i,m-1} - \frac{1}{2}(\tilde{\vartheta}_{i,m}^2$$
$$+ \tilde{\Gamma}_{i,m}^2) + \frac{1}{2}(\vartheta_{i,m}^2 + \Gamma_{i,m}^2) + \sigma_{i,m}(\vartheta_{i,m} + \Gamma_{i,m}). \quad (17)$$

## IV. STABILITY ANALYSIS

*Proof*: We first define a Lyapunov function $V = \sum_{i=1}^N\sum_{k=1}^n V_{i,k}$. It follows that

$$\dot{V} \leq - \sum_{i=1}^N\sum_{k=1}^n c_{i,k}z_{i,k}^2 - \frac{1}{2}\sum_{i=1}^N\sum_{k=1}^n(\tilde{\vartheta}_{i,k}^2 + \tilde{\Gamma}_{i,k}^2) + \mathcal{B}$$
$$\leq - \mathcal{C}V + \mathcal{B}, \quad (18)$$

where $\mathcal{C} = \min\{c_{i,1}, ..., c_{i,n}, \kappa_{i,1}, ..., \kappa_{i,n}, \gamma_{i,1}, ..., \gamma_{i,n}\} > 0$, $\mathcal{B} = \sum_{i=1}^N\sum_{k=1}^n \frac{1}{2}(\vartheta_{i,k}^2 + \Gamma_{i,k}^2) + \sigma_{i,k}(\vartheta_{i,k} + \Gamma_{i,k})$.

We can obtain

$$V(t) \leq V(0)e^{-\mathcal{C}t} + \frac{\mathcal{B}}{\mathcal{C}}(1 - e^{-\mathcal{C}t}), \quad (19)$$

$$\lim_{t\to+\infty}||z_{*,1}(t)|| \leq \sqrt{2V} \leq \sqrt{2V(0) + 2\frac{\mathcal{B}}{\mathcal{C}}}, \quad (20)$$

where $z_{*,1}(t) = [z_{1,1}, z_{2,1}, ..., z_{N,1}]^T$.

The result in (20) illustrates that $e_i$ is bounded and $e_i$ is constrained by performance function (8), which implies that the synchronization error $e_i$ satisfy $-\rho_i^* < e_i < \rho_i^*$ with $t \in [0, \tau_{mp})$ and $-\rho_i \leq -\rho_i^* < e_i < \rho_i^* \leq \rho_i$ with $t \in [\tau_{mp}, +\infty)$. Meanwhile, one has

$$||y - y_r^*|| \leq \frac{||e^*||}{\underline{\Psi}(\mathcal{L} + \mathcal{A})}, \quad (21)$$

where $\underline{\Psi}(\mathcal{L} + \mathcal{A})$ is the minimum singular value of $\mathcal{L} + \mathcal{A}$, $y = [y_{1,1}, y_{2,1}, ..., y_{N,1}]^T \in \mathbb{R}^N$, $y_r^* = [y_r, y_r, ..., y_r]^T \in \mathbb{R}^N$, $e^* = [e_1, e_2, ..., e_N]^T \in \mathbb{R}^N$.

## V. CONCLUSIONS

We propose a performance function reconstruction approach for reconstructing the initial performance function into a new performance function, thereby ensuring that the initial values of the tracking error align with the newly defined performance bounds. Contrasted with existing methods for prescribed performance, the proposed scheme offers two primary advantages. On the one hand, it reduces the overshoot of the tracking error, on the other hand, it ensures that the tracking error converges to the initial performance function bound within the specified time.

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
