# OpenReview forum: "Prescribed Performance-Based Consensus Tracking for Multiagent Systems"
_IEEE.org/ICIST/2024/Conference — IEEE ICIST 2024 Conference Submission_

### Official Review · Reviewer_xjkB · 2024-08-20
**Recommended rejection**

**Rating:** 3
**Confidence:** 4

**Review:**

The innovation points of this manuscript are insufficient and the logic is unclear. It is not recommended to publish this paper in ICIST 2024.

---

### Official Review · Reviewer_CvgH · 2024-08-21
**This paper investigates the prescribed performance-based cooperative control problem for a class of uncertain strict-feedback nonlinear multiagent systems with input saturation. However, the manuscript lacks clarity and organization, making it difficult for readers to understand the proposed control scheme and its implementation.**

**Rating:** 3
**Confidence:** 3

**Review:**

The manuscript does not offer significant advancements over existing techniques or provide a thorough comparative analysis, limiting its contribution to the field. As such, I do not think this paper is suitable for ICIST 2024.

---

### Official Review · Reviewer_c9TY · 2024-08-21
**This paper lacks sufficient innovation, therefore it is recommended to reject the submission.**

**Rating:** 3
**Confidence:** 3

**Review:**

This paper lacks sufficient innovation, therefore it is recommended to reject the submission. The reasons for rejection are as follows：
1.While the newly designed performance function aims to enhance the feasible condition of PPC and accelerate the convergence rate of tracking errors by incorporating a specific logarithmic term, the paper lacks rigorous theoretical proofs to substantiate these claims.  Moreover, the comparison with the performance function outlined in [8] is qualitative and lacks quantitative evaluations through experiments or simulations across multiple scenarios.  Consequently, it is difficult to comprehensively assess the practical value and generality of the proposed function.
2.The novel topology optimization method based on the minimum communication distance, which claims to eliminate redundant edges while ensuring a spanning tree in the communication graph, suffers from inadequate experimental validation. The lack of detailed experimental setups, parameter tuning processes, and test results across varying scenarios hinders the assessment of the method's effectiveness and stability in practical applications. Furthermore, the absence of comparisons with other topology optimization approaches limits a comprehensive evaluation of its novelty.
3. There are many errors in the format of the article and the content is simple, which is not suitable for publication.

---

### Decision · Program_Chairs · 2024-09-08

Reject